# Dimension Reduction Localization Algorithm of Mixed Sources Based on MEMS Vector Hydrophone Array

**DOI:** 10.3390/mi13040626

**Published:** 2022-04-15

**Authors:** Zhenzhen Shang, Libo Yang, Wendong Zhang, Guojun Zhang, Xiaoyong Zhang, Hairong Kou

**Affiliations:** 1Department of Intelligence and Automation, Taiyuan University, Taiyuan 030032, China; b1606004@st.nuc.edu.cn (X.Z.); b1606009@st.nuc.edu.cn (H.K.); 2State Key Laboratory of Dynamic Testing Technology, North University of China, Taiyuan 030051, China; wdzhang@nuc.edu.cn (W.Z.); zhangguojun1977@nuc.edu.cn (G.Z.)

**Keywords:** mixed sources, MUSIC algorithm, MEMS vector hydrophone, dimension reduction, port and starboard ambiguity

## Abstract

In this paper, a mixed sources dimension reduction Multiple Signal Classification (MUSIC) localization algorithm suitable for Micro-Electro-Mechanical System (MEMS) vector hydrophone linear arrays is proposed, which reduces the two-dimensional search to one-dimensional local search. Firstly, the Lagrangian function is constructed by quadratic optimization idea to obtain the estimates of azimuth angles. Secondly, the least square method is utilized for optimal match to obtain the direction-of-arrivals (DOAs) and ranges, and the range parameters are judged in Fresnel zone to obtain the azimuth information of all near-field sources. Finally, find the common DOAs and achieve high-resolution separation of far-field and near-field sources. Simulation and field experiments prove that the proposed algorithm only needs a small number of elements can solve the problem of port and starboard ambiguity, does not need to construct high-order cumulants or multi-dimensional search while the parameters are automatically matched with low computational complexity. This study provides an idea of the engineering application of vector hydrophone.

## 1. Introduction

The coexistence of far-field and near-field sources is an issue that cannot be ignored in array signal processing of passive sonar systems [1,2]. There is a problem of port and starboard ambiguity when scalar hydrophones are located. Vector hydrophone can solve this problem by detecting the vibration-velocity and sound pressure signal simultaneously. MEMS cilia vector hydrophone has better low-frequency detection characteristics and high-sensitivity to sound signals, so it is suitable for underwater low-frequency acoustic target detection and location [3,4]. There have been a large number of mature algorithms suitable for far field sources localization, such as the maximum likelihood estimation algorithm [5], the beam form algorithm [6], and the MUSIC algorithm [7]. In the near-field area of the sound source, DOA and range parameters are taken into consideration simultaneously. And some corresponding algorithms have been proposed, such as the two-dimensional MUSIC method [8], two-stage MUSIC method [9], the higher-order Estimation of Signal Parameters via Rotational Invariance Techniques method (ESPRIT) [10], and noncircular sources [11]. Dakulagi proposed a new DOA estimation algorithm using Nystrom method, which is suitable for far field source target estimation [12].

Numerous methods have been proposed to achieve the localization of mixed sources. Tian et al., proposed the sparse representation algorithm using two cumulant vectors [13]. Liang et al., based on high-order statistics proposed a new two-stage MUSIC algorithm, but the computation is relatively high [14]. Liu proposed the ESPRIT method can automatically pair parameters [15]. Then He et al., presented an oblique-projection MUSIC algorithm with poor resolution due to large aperture loss [16]. Yang estimates the DOA and range parameters successively using sparse recovery techniques [17].

Mixed sources estimate are implemented using nested arrays, and these nested array algorithms require more computation [18,19,20]. Wang et al., presented the based on sparse signal construction method [21,22]. Amir presented a fourth-order spatiotemporal algorithm, which has high computational complexity because of the need to construct two spatiotemporal cumulant matrix [23]. It can be observed that these algorithms are generally applicable to scalar hydrophone array and the port and starboard ambiguity problem has always existed. Huang proposed a localization algorithm based on Discrete Fourier Transform (DFT) and Orthogonal Matching Pursuit (OMP) [24]. Molaei proposed a mixed source estimation algorithm based on ESPRIT with high computational complexity [25]. However, there are few researches on mixed source identification using vector hydrophone array. Shang et al., have studied the vector array rank reduction algorithm, which requires multiple one-dimensional global searches, and the calculation complexity is high [26]. In this paper, the vector hybrid source localization algorithm is further derived and applied.

In this paper, in order to further reduce the computational complexity, we proposed a mixed sources dimension reduction parameter estimation method for MEMS vector hydrophone array. Firstly, the Lagrangian function is constructed by quadratic optimization idea to obtain the estimates of azimuth angles. Secondly, the least square method is utilized for optimal match to obtain the direction-of-arrivals (DOAs) and ranges, and the range parameters are judged in Fresnel zone to obtain the azimuth information of all near-field sources. Finally, find the common DOAs and achieve high-resolution separation of far-field and near-field sources. The two-dimensional search method is simplified to one-dimensional local search. The algorithm makes efficient use of all information in the vector array, does not need multi-dimensional search, does not need to construct high order cumulant. The parameters are automatically matched, and it solves the problem of port and starboard ambiguity.

Throughout the paper, superscripts *T* and *H* represent the transpose, conjugate transpose, respectively. ⊗ represents the Kronecker-product operator, $argmin\left\{\cdot \right\}$ represents the variable value when the objective function takes the minimum value. ${||■||}_{F}$ denotes the Frobenius norm.

## 2. Mixed Far-Field and Near-Field Vector Model

### 2.1. The Work Principle of Composed MEMS Vector Hydrophone

The compound MEMS vector hydrophone is composed of a cilia bionic hydrophone and an acoustic pressure hydrophone, which mimics the sound perception principle of fish lateral line organs. MEMS cilia composed vector hydrophone can realize the simultaneous detection of sound pressure and vibration velocity in the sound field [26]. The structural assembly drawing of the composed vector hydrophone is shown in Figure 1.

The perception principle of fish lateral line organs to acoustic signals are that sound waves through its lateral line holes promote the flow of internal mucus, and it will cause the disturbance of movable cilium in its neural mound. So that the sensory cells around the cilia can obtain stimulation and passed into the fish brain through the nerve tissue. The vector module adopts the bionic principle design idea to imitate the sensing principle of fish lateral line organs [27], and its core sensitive structures are cilia and cross beam. Cilia deflect under the action of sound waves, which then causes the cantilever beam to bend and deform. The resistance value on the beam is changed and then the voltage is output through the Wheatstone bridge. The scalar sensitive unit uses a piezoelectric ceramic tubes to measure the sound pressure signal. The bionic principle of the sensor vector microstructure is shown in Figure 2. As a reference the specific manufacture processes about MEMS chip was revealed in Xue’s paper [4].

### 2.2. The Received Signal Characteristics of Single Vector Hydrophone

In this paper, we consider the two-dimensional MEMS composed vector hydrophone which consists of two vibration velocity channels and a sound pressure channel. $P=p\left(t\right)$ is sound pressure, $v_{x}=v\left(t\right)cos\theta$ and $v_{y}=v\left(t\right)sin\theta$ are mutually perpendicular vibration velocity, θ is the incident angle of signal source, $z=jkr/\left(1+jkr\right)\rho c$ is acoustic impedance coefficient, k is wave number, ρ is medium density and c is the sound velocity. 

Vector channel sensitivity and phase of the vector hydrophone are consistent. The scalar channel has omnidirectional directivity, while the vibration velocity channel has a cosine directivity of “8”. Using this vector directivity characteristic, the port and starboard signal can be distinguished and the vector gain is 3 dB. The gain and beam directivity of the vector hydrophone array are utilized to realize multi-target estimation. The horizontal vector linear array has better gain and monopole directivity, which has guiding significance for the research of location estimation. The relationship between vector linear array gain and beam widths can be expressed as
(1)$$G=10\times logM+3\;\left(\mathrm{dB}\right)$$
(2)$$w_{v}={\left[\frac{sin\left(\frac{M\times \pi \times d}{\lambda }\times cos\left(\theta \right)\right)}{M\times sin\left(\frac{\pi \times d}{\lambda }\times cos\left(\theta \right)\right)}\right]}^{2}\times v_{0}\left(\theta \right)$$
where G is the gain of vector linear array, $w_{v}$ is the beam width of vector linear array, *M* is the number of array elements. λ is the wavelength of the source, d is the array element spacing, $v_{0}\left(\theta \right)$ is the directivity of the vector channel. Therefore, we mainly consider using vector linear array to achieve high precision target positioning.

### 2.3. Mixed Far-Field and Near-Field Signal Array Model

It is assumed that the horizontal uniform linear array of the composed vector hydrophone includes (M=2N+1) vector array elements. K narrow-band uncorrelated signals are simultaneously incident on the array (including far-field and near-field signals). There is not any amplitude and phase error in each array, and mutual coupling interference between arrays is ignored. Where the array noise $N\left(t\right)$ is zero mean Gaussian white noise. The mixed sources model is shown in Figure 3.

It is assumed that the vector matrix is uniformly and symmetrically distributed on the *x*-axis of the rectangular coordinate system. If the *0*-th hydrophone is the reference array element, the received signal of the vector hydrophone linear array can be expressed as
(3)$$X\left(t\right)=\left[a_{v}\left(\theta_{1},r_{1}\right),\ldots ,a_{v}\left(\theta_{K,}r_{K}\right)\right]s\left(t\right)+N\left(t\right)$$

The direction vector of the *i*-th signal can be expressed as
(4)$$a_{v}\left(\theta_{i},r_{i}\right)=a\left(\theta_{i},r_{i}\right)⊗u\left(\theta_{i}\right)$$
where $u\left(\theta_{i}\right)={\left[1\frac{1+\frac{1}{jkr_{i}}}{\rho c}cos\theta_{i}\frac{1+\frac{1}{jkr_{i}}}{\rho c}sin\theta_{i}\right]}^{T}=u\left(\alpha_{i}\right)\approx {\left[1cos\theta_{i}sin\theta_{i}\right]}^{T}$, assuming acoustic impedance coefficient is ρc=1, ignore the effects of orientation and choose the real part information. The direction vector $a\left(\theta_{i},r_{i}\right)$ can be expressed as
(5)$$\left(\theta_{i},r_{i}\right)={\left[e^{i\left(-N\right)\alpha_{i}+{\left(-N\right)}^{2}\beta_{i})},\cdots ,e^{i\omega (N\alpha_{i}+N^{2}\beta_{i})}\right]}^{T}$$
(6)$$\alpha_{i}=\frac{-2\pi d}{\lambda }cos\left(\theta_{i}\right)$$
(7)$$\beta_{i}=\frac{\pi d^{2}}{\lambda r_{i}}sin^{2}\left(\theta_{i}\right)$$
where $\theta_{i}$ and $r_{i}$ are the DOA and range of the *i*-th source, respectively $\theta_{i}\in \left[0,2\pi \right],r_{i}\in [0.62\sqrt{\frac{D^{3}}{\lambda }},+\infty )$. The array aperture is $D=\left(M-1\right)\times d$. When the *i*-th source is a far field source, the range is infinite, then the parameter $\beta_{i}$ tends to zero, and only the DOA information needs to be considered.

In the rest of this article, we assume that the following three assumptions are true:All target signals are independent and narrowband stationary and the noise is the white Gaussian noises;In order to avoid ambiguity estimation, make the array element spacing within a quarter wavelength [28];The number of sound sources must be less than the number of MEMS hydrophones.

## 3. Mixed Sources Reduced-Dimension Location Algorithm for Vector Hydrophone Array

The signal subspace of the data is orthogonal to the noise subspace, that is, the steering vector of the incident signal is orthogonal to the noise subspace [29]. According to the orthogonal characteristics, the array covariance matrix of the received signal can be decomposed into
(8)$$R=E\left(X\times X^{H}\right)=U_{S}\times \underset{S}{\sum }U_{S}^{H}+U_{N}\times \underset{N}{\sum }U_{N}^{H}$$
where R is the array covariance matrix, $\sum_{S}$ and $\sum_{N}$ represent the diagonal matrices composed of signal and noise eigenvalues, respectively.$U_{S}$ represents the signal eigenmatrix corresponding to *K* large eigenvalues, and $U_{N}$ represents the noise eigenmatrix corresponding to $\left(M-K\right)$ small eigenvalues.

Based on the vector rank reduction algorithm [11], further optimization processing is carried out to reduce the computational complexity through the idea of quadratic optimization and the least square method. Next, the optimization algorithm in this paper was introduced in detail.

### 3.1. DOAs Estimate of All Far-Field and Near-Field Sources

Based on the symmetry of the array, Equation (5) can be further transformed and decomposed into the form of the product of two matrices
(9)$$a\left(\theta_{i},r_{i}\right)=\left[\begin{array}{c} \begin{array}{c} e^{j\left(-N\right)\alpha_{i}+{\left(-N\right)}^{2}\beta_{i})}\\ \vdots \end{array}\\ 1\\ \vdots \\ e^{jN\alpha_{i}+N^{2}\beta_{i})} \end{array}\right]=\left[\begin{array}{c} \begin{array}{ccc} e^{j\left(-N\right)\alpha_{i}} & \cdots & 0\\ \vdots & \ddots & \vdots \\ 0 & \cdots & 1 \end{array}\\ \begin{array}{ccc} \;\vdots & ⋰ & \;\vdots \\ e^{j\left(N\right)\alpha_{i}} & \cdots & \;0 \end{array} \end{array}\right]\left[\begin{array}{c} e^{j{\left(-N\right)}^{2}\beta_{i}}\\ e^{j{\left(-N+1\right)}^{2}\beta_{i}}\\ \vdots \\ 1 \end{array}\right]=\zeta \left(\alpha_{i}\right)\eta \left(\beta_{i}\right)$$

In Equation (9), $\zeta \left(\alpha_{i}\right)$ and $\eta \left(\beta_{i}\right)$ can be expressed as
(10)$$\zeta \left(\alpha_{i}\right)=\left[\begin{array}{c} \begin{array}{ccc} e^{j\left(-N\right)\alpha_{i}} & \cdots & 0\\ \vdots & \ddots & \vdots \\ 0 & \cdots & 1 \end{array}\\ \begin{array}{ccc} \vdots & ⋰ & \vdots \\ e^{j\left(N\right)\alpha_{i}} & \cdots & 0 \end{array} \end{array}\right]$$
(11)$$\eta \left(\beta_{i}\right)=\left[\begin{array}{c} e^{j{\left(-N\right)}^{2}\beta_{i}}\\ e^{j{\left(-N+1\right)}^{2}\beta_{i}}\\ \vdots \\ 1 \end{array}\right]$$
where $\zeta \left(\alpha_{i}\right)$ contains only the azimuth information of the sound source, $\eta \left(\beta_{i}\right)$ contains both azimuth information and range information.

Then according to the kronecker product mixed product property we can find that
(12)$$a_{v}\left(\theta_{i},r_{i}\right)=a_{v}\left(\alpha_{i},\beta_{i}\right)=a\left(\theta_{i},r_{i}\right)⊗u\left(\alpha_{i}\right)=\left[\zeta \left(\alpha_{i}\right)\eta \left(\beta_{i}\right)\right]⊗u\left(\alpha_{i}\right)=\left[\zeta \left(\alpha_{i}\right)⊗u\left(\alpha_{i}\right)\right]\eta \left(\beta_{i}\right)=\nu \left(\alpha_{i}\right)\eta \left(\beta_{i}\right)$$

So the spatial spectral function of Equation (10) can be expressed as
(13)$$P\left(\theta ,r\right)=\frac{1}{\eta^{H}\left(\beta_{i}\right)\nu^{H}\left(\alpha_{i}\right)U_{N}U_{N}{}^{H}\nu \left(\alpha_{i}\right)\eta \left(\beta_{i}\right)}=\frac{1}{\eta^{H}\left(\beta_{i}\right)UU\left(\alpha_{i}\right)\eta \left(\beta_{i}\right)}$$

The above problem can be evolved to solve the quadratic optimization problem, the implicit constraint is as in Equation (14).
(14)$$\left\{\begin{array}{c} e^{H}\eta \left(\beta_{i}\right)=1\\ e={\left[0,\cdots ,0,1\right]}^{T}\in R^{\left(N+1\right)\times 1} \end{array}\right.$$

Construct the quadratic optimization problem function as follows:(15)$$ff_{min}=min\eta^{H}\left(\beta_{i}\right)UU\left(\alpha_{i}\right)\eta \left(\beta_{i}\right)s.t.e^{H}\eta \left(\beta_{i}\right)=1$$

Using the Lagrange operator method, the operator parameter λ is introduced to construct the cost function of the quadratic optimization problem.
(16)$$L\left(\alpha_{i},\beta_{i}\right)=\eta^{H}\left(\beta_{i}\right)UU\left(\alpha_{i}\right)\eta \left(\beta_{i}\right)-\lambda \left(e^{H}\eta \left(\beta_{i}\right)-1\right)$$

The partial derivative of $L\left(\alpha_{i},\beta_{i}\right)$ versus $\eta \left(\beta_{i}\right)$ can expressed as
(17)$$\frac{\partial L\left(\alpha_{i},\beta_{i}\right)}{\partial \eta \left(\beta_{i}\right)}=2UU\left(\alpha_{i}\right)\eta \left(\beta_{i}\right)-\lambda e=0$$
(18)$$\eta \left(\beta_{i}\right)=\lambda_{0}UU^{-1}\left(\alpha_{i}\right)e$$

According to the Equations (14) and (18), we can obtain
(19)$$\lambda_{0}=\frac{1}{e^{H}\cdot UU^{-1}\left(\alpha_{i}\right)\cdot e}$$
(20)$$\eta \left(\beta_{i}\right)=\frac{UU^{-1}\left(\alpha_{i}\right)\cdot e}{e^{H}\cdot UU^{-1}\left(\alpha_{i}\right)\cdot e}$$

From the above derivation, the estimated value of the intermediate parameter is
(21)$$\hat{\alpha_{i}}=argmin\frac{1}{e^{H}\cdot UU^{-1}\left(\alpha_{i}\right)\cdot e}=argmaxe^{H}\cdot UU^{-1}\left(\alpha_{i}\right)\cdot e$$

Because $\alpha_{i}$ is normalized to $\bar{\alpha_{i}}=cos\theta_{i}$, and one dimensional local search in the range of $\left[-1,\;1\right]$ can obtain $\bar{K}$ peak points, which corresponds to $\bar{K}$ parameter values $\hat{\alpha_{i}}$. All estimates of azimuth parameters for far-field and near-field mixed sources are obtained [26].
(22)$$\hat{\theta_{i}}=arccos\left(\frac{-2\pi d}{\lambda \alpha_{i}}\right)$$

At the same time, we can notice that in real situations, some far-field sources have the same DOA as near-field sources. That is, the estimated value $\bar{K}$ will not exceed the actual number of sources K. When all the sound sources have different azimuth angles, we have $\bar{K}=K$.

### 3.2. The Range Estimate of All Far-Field and Near-Field Sources

Next, we estimate the range parameters of mixed sources. The Equation (11) can be converted to
(23)$$\eta_{1}\left(\beta_{i}\right)=\left[\begin{array}{c} 1\\ e^{j\beta_{i}}\\ \vdots \\ e^{j{\left(-N\right)}^{2}\beta_{i}} \end{array}\right]$$

Then the phase Angle can be expressed as
(24)$$\hat{\varphi_{i}}=angle\left[\eta_{1}\left(\beta_{i}\right)\right]={\left[\begin{array}{c} 0\\ \beta_{i}\\ \vdots \\ {\left(-N\right)}^{2}\beta_{i} \end{array}\right]}_{\left(N+1\right)\times 1}=\left[\begin{array}{c} 0\\ {\left(-1\right)}^{2}\\ \vdots \\ {\left(-N\right)}^{2} \end{array}\right]\beta_{i}=p\beta_{i}$$

Find the optimal function matching value by least square method
(25)$$min||q{∆}_{i}-{\hat{\varphi }}_{i}{||}_{F}^{2}$$
where $q=\left[1_{N+1},p\right]={\left[\begin{array}{c} 1\\ 1\\ \vdots \\ 1 \end{array}\begin{array}{c} 0\\ {\left(-1\right)}^{2}\\ \vdots \\ {\left(-N\right)}^{2} \end{array}\right]}_{\left(N+1\right)\times 2},{∆}_{i}={\left[{∆}_{i0},\hat{\beta_{i}}\right]}^{T}\in R^{2\times 1},{∆}_{i0}$ is the parameter estimation error value.

Least squares function as the solution of ${∆}_{i}$ can be expressed as
(26)$${∆}_{i}={\left(q^{T}q\right)}^{-1}q^{T}\hat{\varphi_{i}}$$

At the same time parameter $\hat{\alpha_{i}}\;\mathrm{and}\;\hat{\beta_{i}}$ are automatic matching, then the range from the source mentioned above can be parameter estimates
(27)$$\hat{r_{i}}=\frac{\pi d^{2}}{\lambda \hat{\beta_{i}}}sin^{2}\hat{\theta_{i}}$$

By judging the range parameter in the near-field Fresnel zone, the range value corresponding to the near-field azimuth is obtained.
(28)$$\left\{\begin{array}{c} \hat{r_{i}}\in \left(0.62\sqrt{\frac{D^{3}}{\lambda }},\frac{2D^{2}}{\lambda }\right),K_{2}\left(nearsources\right)\\ \hat{r_{i}}\in \left(\frac{2D^{2}}{\lambda },\infty \right),K_{1}\left(farsources\right) \end{array}\right.$$

### 3.3. The Range Estimate of All Far-Field and Near-Field Sources

Suppose there are $K_{3}=K-\bar{K}$ common azimuths, and bring $K_{2}$ near field azimuths into the far field subspace spectral function. In general, the azimuth spectrum amplitude of the far-field and near-fields will differ by more than an order of magnitude. Then when the near field azimuth is brought into the Equation (29), the source with a higher spectral value can be regarded as the common azimuth of the far-field and near-fields.
(29)$$P\left(\theta \right)=\frac{1}{\nu^{H}\left(\alpha_{i}\right)U_{N}U_{N}{}^{H}\nu \left(\alpha_{i}\right)}$$

In summary, all the far-field and near-field azimuth information can be obtained to achieve accurate high-resolution estimation in the case of mixed sources.

### 3.4. The Computational Complexity Analysis of Proposed Algorithm

The proposed algorithm shares many advantages, such as does not require multi-dimensional search, does not need to construct high-order cumulants, and the parameters are automatically matched. The dimension reduction transformation from two-dimensional parameter estimation to one-dimensional parameter estimation is realized, and the calculation amount is low. The computations major involve array eigen-decomposition, covariance matrix, one-dimensional local search and common azimuth identification. The total computational complexity can express as
(30)$$O\left\{\begin{array}{c} {\left(3M\right)}^{2}J+{\left(3M\right)}^{3}+\left(3M+1\right)\left(3M-K\right)K_{2}\\ +n_{\theta }\left[\left(3M-K\right)\left(3N+1\right)\left(3M+3N+1\right)+{\left(3N+1\right)}^{3}\right] \end{array}\right\}$$
where the number of snapshots is J, $n_{\theta }$ is the number of DOA peaks searches in the interval. While oblique projection MUSIC needs the overlapping sub-matrix and its eigen-decomposition process, two-stage MUSIC and reduced rank MUSIC algorithm requires multiple global searches. The proposed algorithm requires less computational complexity. Table 1 lists the time complexity of other algorithms. $K_{2}$ is the number of far sources.

## 4. Simulation of Vector Dimension Reduction Localization Algorithm

In this part, we verify the performance of the proposed algorithm through simulation and field experiment. Without loss of generality, suppose a 9-element vector hydrophone linear array with an array spacing of 1/4 wavelength. It is presumed that all array elements have no amplitude and phase consistency errors and there is no mutual coupling reaction. The symbol SNR represent the signal to noise ratio, NS represent the number of snapshots and RMSE represent the root mean square error [30]. A total of 400 Monte Carlo experiments were carried out, and the Cramer Rao-Bound (CRB) is the lower bound of the azimuth estimate [17,31,32]. Comparing the calculation results of this paper with the scalar algorithm, and the advantages of the vector algorithm are proved. The proposed algorithm is compared with the reduced rank algorithm (RRM) and the two-stage MUSIC (TSM) method, which shows the advantages of the proposed algorithm.

### 4.1. The Computational Complexity Analysis of Proposed Algorithm

In this simulation experiment, this algorithm is utilized to compare the difference between scalar and vector array element structure. Supposing that there are three far-field sources, and the DOAs are $\left\{\theta_{1}=10{}^{\circ},\theta_{2}=25{}^{\circ},\theta_{3}=60{}^{\circ}\right\}$. We assume NS and SNR are 2000 and 20 dB, respectively. 

Figure 4 is the simulation comparison of scalar and vector array results, which clearly shows that the proposed algorithm achieves accurate estimation of three targets; and meanwhile the scalar algorithm has six target signals, three of which are real targets, and the other three azimuth angles are false targets, they are the symmetrical angles of the three real targets at the port and port positions. The simulation results prove that the vector algorithm resolves the problem of port and starboard ambiguity within the range of $\left[0,\;2\pi \right]$, which is more advantageous than the scalar array algorithm.

### 4.2. The Mixed Far-Field and Near-Field Sources Estimation

This simulation is utilized to verify the performance of the algorithm under mixed far-field and near-field sources. Assuming that two far-field sources and two near-field source have the following characteristics, $\left\{\theta_{1}=10{}^{\circ}\right\}$, $\left\{\theta_{2}=25{}^{\circ}\right\}$, $\left\{\theta_{3}=25{}^{\circ},\lambda_{3}=5\lambda \right\}$, $\left\{\theta_{4}=60{}^{\circ},\lambda_{4}=3\lambda \right\}$. The far-field source and the near-field source have a common azimuth angle of 25°.

First, we assume the NS is 2000 and the SNR is 20 dB. Figure 5a show the calculation results of DOA and range of the mixed source. Judging all the range parameters in the Fresnel zone, we get a far-field DOA of 10° and two near-field DOAs of 25° and 60°. Then all the near-field DOAs are substituted into the Formula (31). The spatial spectrum comparison graph is shown in Figure 5b, where the spectrum value corresponding to 25° azimuth is much larger than the spectrum value corresponding to 60° azimuth. So it can be seen that an azimuth angle of 25° is the common source. Secondly, the NS is set to 2000 to study the effect of SNR on positioning. And the SNR of the three signal sources is increased from −15 dB to 20 dB according to the regularity of 5 dB interval. Figure 6a shows the relationship between the DOA estimation accuracy and the SNR. And Figure 6b shows the relationship between the DOA estimation accuracy and the NS. Third, assuming that the SNR is 20 dB, we study the impact of the NS on positioning. And NS changes from 100 to 2000 in increments of 100. Figure 7a,b show the influence of SNR and NS on the accuracy of range estimation. The comparison results of the rank reduction algorithm and the second-order MUSIC algorithm and CRB are also shown in the figure.

As the SNR and NS increases, the RMSE of DOA estimation and range estimation decreases. The proposed algorithm has better performance.

### 4.3. The Field Experiment of Mixed Sources Estimation Algorithm

The field experiment was carried out in a reservoir with an average water depth of 30 m, which is a good field test environment. The test site is shown in Figure 8, where the source and the hydrophone are placed 10 m underwater. The vector hydrophone linear array is composed of 5-element array whose array spacing is a quarter of a wavelength, and two fish-lip emitting transducers emit narrowband signals of 500 Hz and 800 Hz, respectively. During the experiment, the hydrophone array was fixed on the floating dock, the source No. 1 was hoisted from the floating dock, and the source No. 2 was hoisted underwater by a tugboat at the center of the lake. The vector hydrophone array is placed in the near-field of source No. 1 and the far-field of source No. 2. We use the NI acquisition card to collect the signals received and estimate the orientation of multiple targets.

Utilize the proposed algorithm to estimate the azimuth of the far-field and the near-field sources. Figure 9 shows the estimated azimuth angles of all sources. Figure 10 shows the azimuth and range diagrams of the far and near-field sound sources, and the red asterisk represents the location of the target. The calculated results are consistent with the real GPS data, as can be seen that the azimuth of the near-field sound source is (46°, 2*λ*), and the DOA of far-field source is 284°. The field experimental results verify the performance of the mixed source algorithm, which has an important guiding role in the engineering application of vector hydrophones.

## 5. Conclusions

In this paper, a mixed sources dimension reduction MUSIC algorithm suitable for linear arrays of vector hydrophones is proposed, which reduces the two-dimensional search method to one-dimensional local search. The signal direction vector is expressed as the product of two parameter matrixes with independent azimuth and range parameters. The Lagrangian function is constructed using the quadratic optimization idea to obtain all azimuth angles; then the least square method is used to automatically match to obtain the azimuth angles and ranges. The range parameters are judged in the Fresnel zone to obtain the azimuth information of all near field sources; the common azimuth angle of the far-field and near-fields is identified, and finally the high-resolution separation of all the far-field and near-field sources is realized. The proposed algorithm solves the problem of port and starboard ambiguity and it does not need multi-dimensional search or to construct high-order cumulants while the parameters are automatically matched with low computational complexity. The above shows that MEMS vector hydrophone has broad prospects in the field of underwater acoustic detection.

## Figures and Tables

**Figure 1 micromachines-13-00626-f001:**
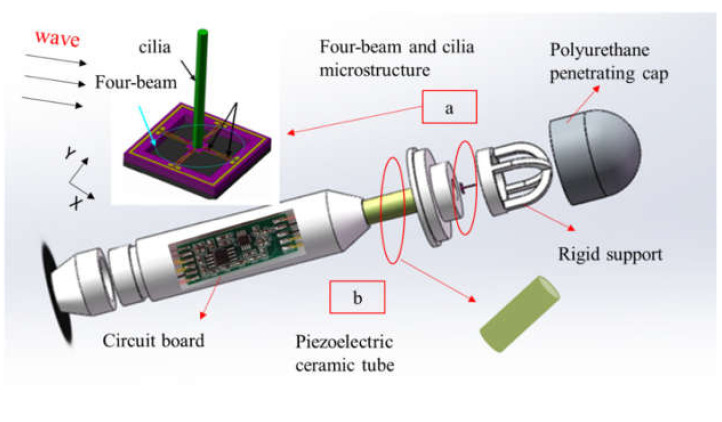
The assembly drawing of vector hydrophone. (**a**) four-beam and cilia microstructure; (**b**) piezoelectric ceramic tube.

**Figure 2 micromachines-13-00626-f002:**
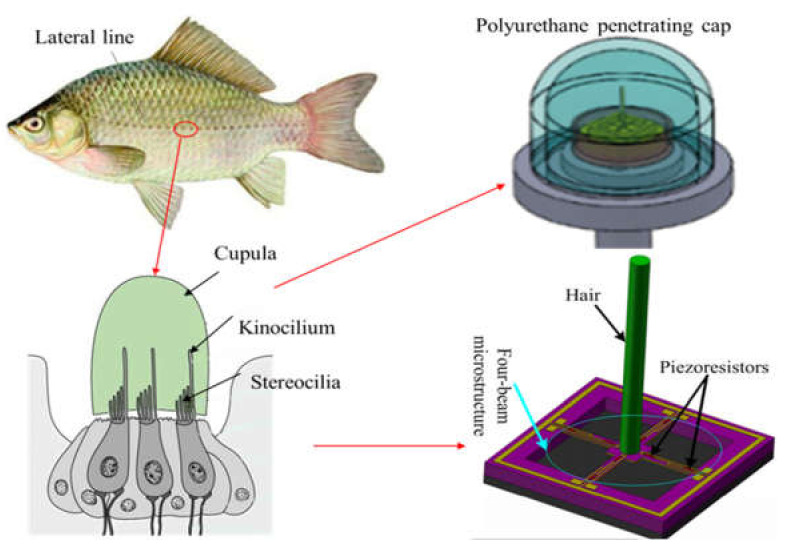
The bionic principles and micro-structural models.

**Figure 3 micromachines-13-00626-f003:**
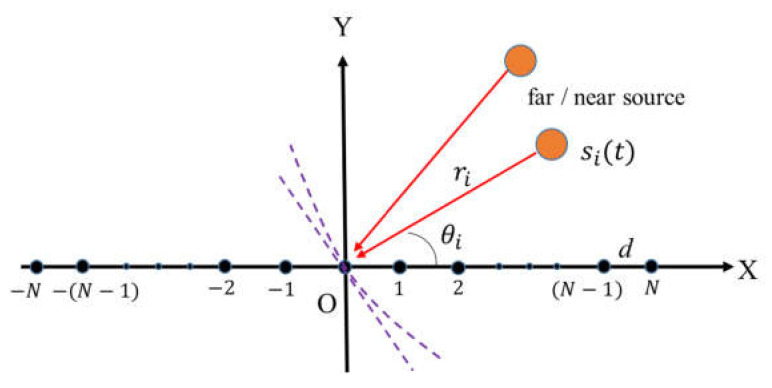
Mixed sources signal model of vector hydrophone.

**Figure 4 micromachines-13-00626-f004:**
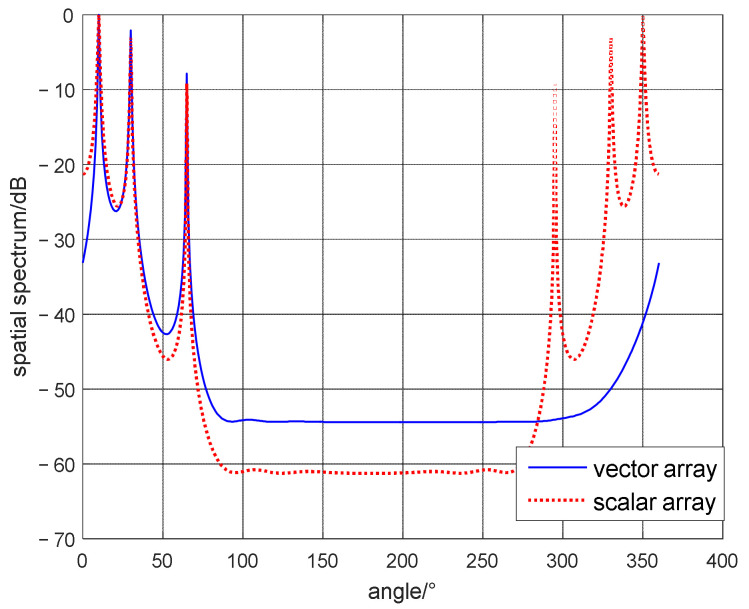
The different of scalar and vector array localization algorithm.

**Figure 5 micromachines-13-00626-f005:**
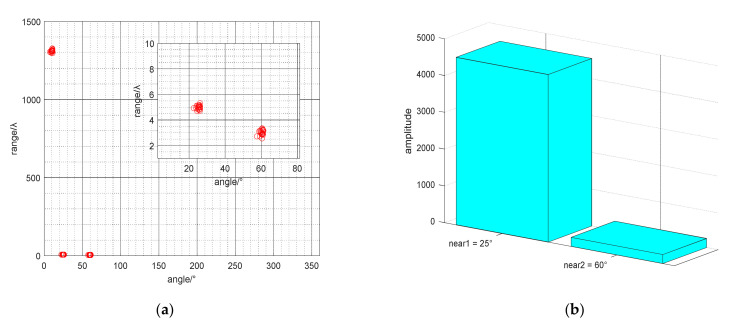
(**a**) The scatter diagram of DOA and range estimation; (**b**) The spectral comparison graph about common DOAs.

**Figure 6 micromachines-13-00626-f006:**
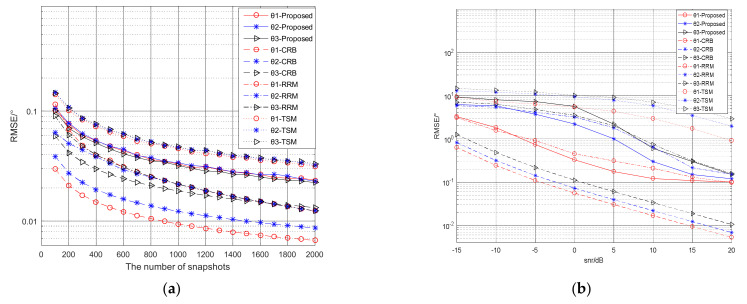
(**a**) The relationship between NS and the RMSE of the DOAs estimation; (**b**) The relationship between the SNR and the RMSE of the DOAs estimation.

**Figure 7 micromachines-13-00626-f007:**
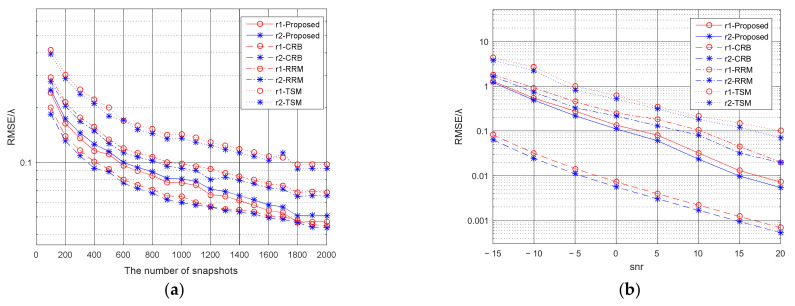
(**a**) The relationship between NS and the RMSE of the ranges estimation; (**b**) The relationship between the SNR and the RMSE of the ranges estimation.

**Figure 8 micromachines-13-00626-f008:**
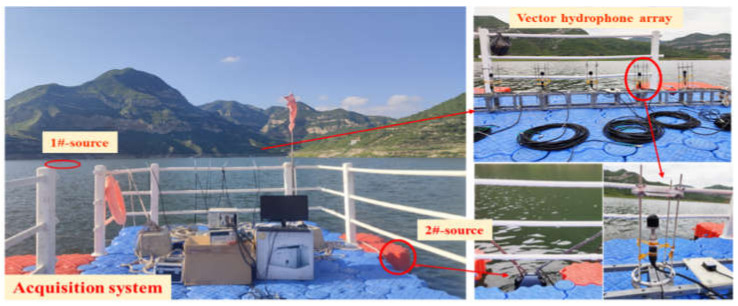
The field test map and hydrophone array distribution map.

**Figure 9 micromachines-13-00626-f009:**
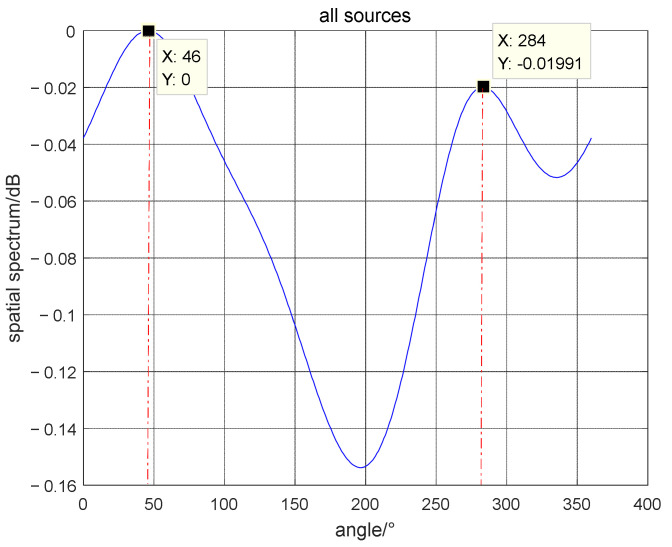
The azimuth of all sound sources.

**Figure 10 micromachines-13-00626-f010:**
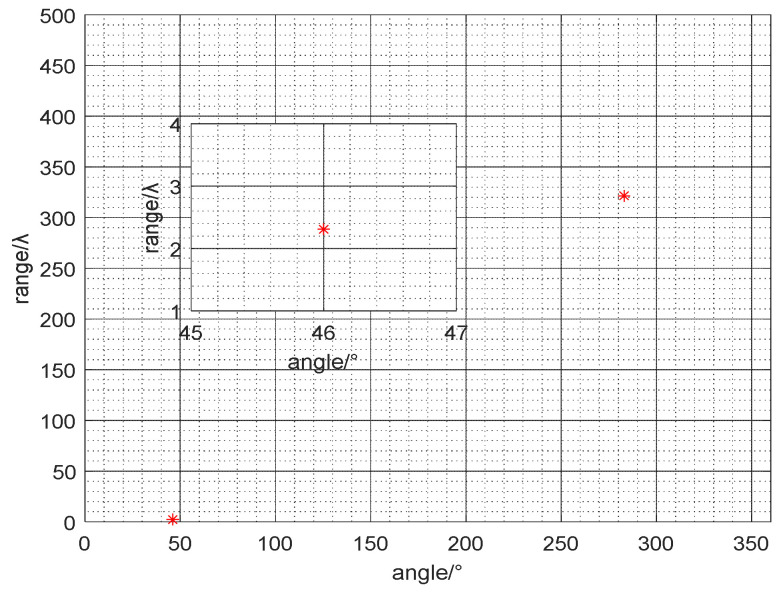
The azimuth map of mixed sources (Red asterisk represents the location of the target).

**Table 1 micromachines-13-00626-t001:** Computational complexity of different algorithms.

Algorithm	Computational Complexity
RRM	$O\left\{\begin{array}{c} {\left(3M\right)}^{2}J+{\left(3M\right)}^{3}+\left(3M+1\right)\left(3M-K\right)K_{2}\\ +n_{\theta }\left[\left(3M-K\right)\left(3N+1\right)\left(3M+3N+1\right)+{\left(3N+1\right)}^{3}\right]\\ +n_{r}\left(3M-K\right)\left(3M+1\right)+n_{r}K_{2}\left(3M-K\right)\left(3M+1\right) \end{array}\right\}$
TSM	$O\left\{{\left(3M\right)}^{2}J+{\left(3M\right)}^{3}+n_{\theta }n_{r}\left(3M-K\right)\left(3N+1\right)\left(3M+3N-1\right)\right\}$
Proposed	$O\left\{\begin{array}{c} {\left(3M\right)}^{2}J+{\left(3M\right)}^{3}+\left(3M+1\right)\left(3M-K\right)K_{2}\\ +n_{\theta }\left[\left(3M-K\right)\left(3N+1\right)\left(3M+3N+1\right)+{\left(3N+1\right)}^{3}\right] \end{array}\right\}$

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
