# Peer review of "Dimension Reduction Localization Algorithm of Mixed Sources Based on MEMS Vector Hydrophone Array"

_micromachines, 2022, doi:10.3390/mi13040626_

Round 1

Reviewer 1 Report

This paper addresses the optimization algorithm to reduce the dimension in the localization application of vector hydrophone arrays. Through sufficient theoretical derivation, the vector hydrophone array shows a high performance no matter in simulation or reservoir field. However, there’re lots of mistakes in this paper. The problems are listed as follows:

Grammar mistakes and improper expression:

Page 5 Line 153 “Further optimize the algorithm … the lowest square method”

Page 6 Line 165 “Formula 10”

Page 7 Line 175 “Because … so…”

Please ask a native speaker or editing service to check the writing.

Page 9 line 229 “SNR, NS and RMSE”. The acronyms should be spelled in full when first appearing.

Page 5 Line 135 “following four assumptions”. But there are only three assumptions.

The description on equations, formulas, and figures:

All the equations and formulas need to be described more clearly. All the variable names on the left side of the equation should be explained.

For example, what is “G” in equation (1), “N(t)” in equation (3), and R in equation (8)?

In equation (4), u(θi) is the same as u(θi)?

Page 4 Line 115, what’s the meaning of “N”?

The relationship between the figures and text should be strengthened. For example, Figures 1 and 2 are the structure and principle of the hydrophone. Without enough description, readers can hardly understand these.

Specifically, Figures 6 and 7 are the key results of the experiment. However, the descriptions are quite a few. As a result, the author should focus more on figure descriptions.

Analysis of computational complexity

The key point of the proposed algorithm is to reduce computational cost and the authors make effort to emphasize it on Page 2 Line 44 to Line 66. However, the authors should list the time complexity function of the other algorithms or describe why the computational cost is high, not just mention the computational cost is high.

On the other hand, section 3.4 (analysis on computational complexity) needs more descriptions. Besides, can the equation (30) be simplified or not?

To illustrate the advantage of the proposed algorithm, a comparison with other algorithms is necessary as well. Please add some comparisons in the result part.

Other comments:

Page 2 Line 67 to 73, the authors should put more emphasis on the contributions instead of the work report.

Page3 Line 106 to 108, the logic is not proper to support the conclusion. 

Author Response

Dear Editor:

Sorry for disturbing you. This paper has been submitted to the “Micromachines” once (micromachines-1630138), and advised by two reviewers. There are some modifications occurred according to the comments. We submit here the revised manuscript as well as a list of changes.

Enclosed here please find a manuscript authored by Zhenzhen Shang, Libo Yang, Wendong Zhang, Guojun Zhang, Xiaoyong Zhang and Hairong Kou and entitled “Dimension Reduction Localization Algorithm of Mixed sources Based on MEMS Vector Hydrophone Array”. In this paper, in order to further reduce the computational complexity, we proposed a mixed sources dimension reduction parameter estimation method for MEMS vector hydrophone array. The two-dimensional search method is simplified to one-dimensional local search. The proposed algorithm further reduces the computational complexity, and realizes automatic parameter matching through the idea of secondary optimization and least square method. Simulation and field experiments prove that the proposed algorithm only needs a small number of array elements can solve the problem of port and starboard ambiguity, does not need to construct high-order cumulants or multi-dimensional search while the parameters are automatically matched with low computational complexity. This study provides an idea of the engineering application of vector hydrophone.

Thank you and look forward to having your opinion.

Correspondence and phone calls about the paper should be directed to Zhenzhen Shang at phone and fax number, and e-mail address:

Phone: 86-18235139505;

Fax: 86-0351-3922131;

Sincerely yours,

Zhenzhen Shang

Department of intelligence and automation, Taiyuan University.

Taiyuan 030032, China.

Reviewer 2 Report

Comments and Suggestions for Authors

This study relates linear arrays of vector hydrophones for reducing two-dimensional search to one-dimensional local search. The author used the square method for optimizing azimuth angles and identifying sources of near-field and far-field. The proposed design and optimization are solving the problem of port and starboard ambiguity. The author also carried out a field experiment on the mixed source. This paper can be accepted subject to some minor revisions as below.

  1. “Amir presented fourth-order spatiotemporal algorithms...” There is missing citation information in the reference lists. (Row 54 on page 2) Please refer to it.
  2. In order to be clear for the reader, the author needs to explain the K value or cite a related reference in row 176 of page 7. How direction of arrivals (DOA) from the near-field source the same as the far-field source?
  3. Needs a more detailed explanation of the formula (29). How is the source with a higher spectral value regarded as a common azimuth of the far-field and near-field?

Author Response

(The authors gave the same response as above.)

Round 2

Reviewer 1 Report

The authors addressed my questions carefully.